# Posterior Concentration for Sparse Deep Learning

**Nicholas G. Polson and Veronika Ročková**
Booth School of Business
University of Chicago
Chicago, IL 60637

## Abstract

We introduce Spike-and-Slab Deep Learning (SS-DL), a fully Bayesian alternative to dropout for improving generalizability of deep ReLU networks. This new type of regularization enables provable recovery of smooth input-output maps with *unknown* levels of smoothness. Indeed, we show that the posterior distribution concentrates at the near minimax rate for $\alpha$-Hölder smooth maps, performing as well as if we knew the smoothness level $\alpha$ ahead of time. Our result sheds light on architecture design for deep neural networks, namely the choice of depth, width and sparsity level. These network attributes typically depend on unknown smoothness in order to be optimal. We obviate this constraint with the fully Bayes construction. As an aside, we show that SS-DL does not overfit in the sense that the posterior concentrates on smaller networks with fewer (up to the optimal number of) nodes and links. Our results provide new theoretical justifications for deep ReLU networks from a Bayesian point of view.

## 1 Introduction

Deep learning constructs are powerful tools for pattern matching and prediction. Their empirical success has been accompanied by a number of theoretical developments addressing (a) why and when neural networks generalize well, (b) when do deep networks out-perform shallow ones and (c) which activation functions and with how many layers. Despite the flurry of research activity, there are still many theoretical gaps in understanding why deep neural networks work so well. In this paper, we provide several new insights by studying the speed of posterior concentration around the optimal predictor, and in doing so we make a contribution to the Bayesian literature on deep learning rates.

Bayesian non-parametric methods are proliferating rapidly in statistics and machine learning, but their theoretical study has not yet kept pace with their application. Lee (2000) showed consistency of posterior distributions over single-layer sigmoidal neural networks. Our contribution builds on this work in three fundamental aspects: (a) we focus on *deep* rather than single-layer, (b) we focus on *rectified linear units* (ReLU) rather than sigmoidal squashing functions, (c) deploying $\ell_0$ regularization, we show that the posterior converges at an optimal speed beyond the mere fact that it is consistent. To achieve these goals, we adopt a statistical perspective on deep learning through the lens of non-parametric regression.

Using deep versus shallow networks can be justified theoretically in many ways. First, while both shallow and deep neural networks (NNs) are universal approximators (i.e. can approximate any continuous multivariate function arbitrarily well on a compact domain), Mhaskar et al. (2017) show that deep nets use exponentially fewer number of parameters to achieve the same level of approximation accuracy for compositional functions. Kolmogorov (1963) provided another motivation for deep networks by showing that superpositions of univariate semi-affine functions provide a universal basis for representing multivariate functions. Telgarsky (2016) provides examples of functions that cannot be represented efficiently with shallow networks and Kawaguchi et al (2017) explains why deep networks generalize well. In related work, Poggio et al. (2017) show how deep networks can

avoid the curse of dimensionality for compositional functions. These theoretical results are growing and our goal is to show how they can be leveraged to show posterior concentration rates for deep learning. In particular, we will build on properties of deep ReLU networks characterized recently by Schmidt-Hieber (2017).

Deep ReLU activating functions can also be justified by theory. Evidence exists that training deep learning proceeds best when the neurons are either off or operate in a linear way. For example, Glorot et al. (2011) show that ReLU functions outperform hyperbolic tangent or sigmoid squashing functions, both in terms of statistical and computational performance. The success of ReLUs has been partially attributed to their ability to avoid vanishing gradients and their expressibility. The attractive approximation properties of ReLUs were discussed by many authors including Telgarsky (2017), Vitushkin (1964) or Montufar et al. (2014). Schmidt-Hieber (2017) points out a very curious aspect of ReLU activators that their composition can yield rate-optimal reconstructions of smooth functions of an arbitrary order, not only up to order 2 which would be expected from piecewise linear approximators.

It is commonly perceived (Goodfellow et al., 2016) that generalizability of neural networks can be improved with regularization. Regularization, loosely defined as any modification to a learning algorithm that is intended to reduce its test error but not its training error (Goodfellow et al., 2016), can be achieved in many different ways. Beyond ReLU activators, another way to regularize is by adding noise to the learning process. For example, the dropout regularization (Srivastava et al., 2014) samples from (and averages over) thinned networks obtained by randomly dropping out nodes together with their connections. While motivated as stochastic regularization, dropout can be regarded as deterministic $\ell_2$ regularization obtained by margining out dropout noise (Wager, 2014). Dropout averaging over sparse architectures pertains, at least conceptually, to Bayesian model averaging under spike-and-slab priors. Spike-and-slab approaches assign a prior distribution over sparsity patterns (models) and perform model averaging with posterior model probabilities as weights (George and McCulloch, 1993). Similarly to dropout, the spike-and-slab approach effectively switches off model coefficients. However, dropout averages out patterns using equal weights rather than posterior model probabilities. Our approach embeds $\ell_0$ regularization within the layers of deep learning and capitalizes on its connection to Bayesian subset selection. We exploit spike-and-slab constructions not necessarily as a tool for model selection, but rather as a fully Bayesian alternative to dropout in order to (a) inject sparsity in deep learning to build stable network architectures, (b) achieve adaptation to the unknown aspects of the regression function in order to achieve near-minimax performance for estimating smooth regression surfaces.

Casting deep ReLU networks with $\ell_0$ penalization as a Bayesian hierarchical model, we study the speed of posterior convergence around Hölder smooth regression functions. Our first result states that, with properly chosen width, depth and sparsity level, the convergence rate is near minimax optimal when the smoothness is known. Going further, we show that adaptation to smoothness can be achieved by assigning suitable complexity priors over the network width and sparsity.

The rest of the paper is outlined as follows. Section 2 describes our statistical framework for analyzing deep learning predictors. Section 3 defines deep ReLU networks. Section 4 constructs an appropriate spike-and-slab regularization for deep learning. Section 5 contains posterior concentration results for sparse deep ReLU networks. Finally, Section 6 concludes with a discussion.

## 1.1 Notation

The $\varepsilon$-covering number of a set $\Omega$ for a semimetric $d$, denoted by $\mathcal{E}(\varepsilon; \Omega; d)$, is the minimal number of $d$-balls of radius $\varepsilon$ needed to cover set $\Omega$. The notation $a_n \lesssim b_n$ will be used to denote inequality up to a constant, where $a_n \asymp b_n$ if $a_n \lesssim b_n$ and $b_n \lesssim a_n$. The symbol $\lfloor a \rfloor$ denotes the greatest integer that is smaller than or equal to $a$, $\propto$ is equality up to a constant and $\|f\|_\infty$ is the supremum of a function $f$.

## 2 Deep Learning: A Statistical Framework

Deep Learning, in its simplest form, reconstructs high-dimensional input-output mappings. To fix notation, let $Y \in \mathbb{R}$ denote a (low dimensional) output and $\boldsymbol{x} = (x_1, \ldots, x_p)' \in [0, 1]^p$ a (high dimensional) set of inputs.

From a machine learning viewpoint, predicting an outcome from a set of features is typically framed as noise-less non-parametric regression for recovering $f_0 : [0, 1]^p \rightarrow \mathbb{R}$. Given inputs $\boldsymbol{x}_i$ of training data and outputs $Y_i = f_0(\boldsymbol{x}_i)$ for $1 \leq i \leq n$, the goal is to learn a deep learning architecture $\widehat{f}_{\boldsymbol{B}}^{DL}$ such that $\widehat{f}_{\boldsymbol{B}}^{DL}(\boldsymbol{x}) \approx f_0(\boldsymbol{x})$ for $\boldsymbol{x} \notin \{\boldsymbol{x}_i\}_{i=1}^n$. Training neural networks is then positioned as an optimization problem for finding values $\widehat{\boldsymbol{B}} \in \mathbb{R}^T$ that minimize empirical risk ($L^2$-recovery error on training data) together with a regularization term, i.e.

$$\widehat{\boldsymbol{B}} = \arg \min_{\boldsymbol{B}} \sum_{i=1}^n [f_0(\boldsymbol{x}_i) - f_{\boldsymbol{B}}^{\mathrm{DL}}(\boldsymbol{x}_i)]^2 + \phi(\boldsymbol{B}) \qquad (1)$$

where $\phi(\boldsymbol{B})$ is a penalty over the weights and offset parameters $\boldsymbol{B}$. In practice, this is most often carried out with some form of stochastic gradient descent (SGD) (see e.g. Polson and Sokolov (2017) for an overview).

From a statistical viewpoint, deep learning is often embedded within non-parametric regression where responses are linked to fixed predictors in a stochastic fashion through

$$Y_i = f_0(\boldsymbol{x}_i) + \varepsilon_i, \quad \varepsilon_i \overset{iid}{\sim} \mathcal{N}(0, 1), \quad 1 \leq i \leq n. \qquad (2)$$

We define by $\mathcal{H}_p^\alpha = \{f : [0, 1]^p \rightarrow \mathbb{R}; \|f\|_{\mathcal{H}^\alpha} < \infty\}$ the class of $\alpha$-Hölder smooth functions on a unit cube $[0, 1]^p$ for some $\alpha > 0$, where $\|f\|_{\mathcal{H}^\alpha}$ is the Hölder norm. The true generative model, giving rise to (2), will be denoted with $\mathbb{P}_{f_0}^{(n)}$. Assuming $f_0 \in \mathcal{H}_p^\alpha$, we want to reconstruct $f_0$ with $\widehat{f}_{\boldsymbol{B}}^{DL}$ so that the empirical $L^2$ distance

$$\|\widehat{f}_{\boldsymbol{B}}^{DL} - f_0\|_n^2 = \frac{1}{n} \sum_{i=1}^n [\widehat{f}_{\boldsymbol{B}}^{DL}(\boldsymbol{x}_i) - f_0(\boldsymbol{x}_i)]^2$$

is at most a constant multiple away from the minimax rate $\varepsilon_n = n^{-\alpha/(2\alpha+p)}$ (up to a log factor). Unlike related statistical developments (Schmidt-Hieber (2017), Bauer and Kohler (2017)), we approach the reconstruction problem from a purely Bayesian point of view. While the optimization problem (1) has a Bayesian interpretation as MAP estimation under regularization priors, here we study the behavior of the *entire posterior*, not just its mode.

Our approach rests on careful constructions of prior distributions $\pi(f_{\boldsymbol{B}}^{DL})$ over deep learning architectures. In Bayesian non-parametrics, the quality of priors can be often quantified with the speed at which the posterior distribution shrinks around the true regression function as $n \rightarrow \infty$. Ideally, most of the posterior mass should be concentrated in a ball centered around the true value $f_0$ with a radius proportional to the minimax rate $\varepsilon_n$. These statements are ultimately framed in a frequentist way, describing the typical behavior of the posterior under the true generative model $\mathbb{P}_{f_0}^{(n)}$.

In the construction of deep learning priors, a few questions emerge. How does one choose the architecture $f_{\boldsymbol{B}}^{DL}$: how deep and what activation functions? The choice typically depends on how quickly one can reconstruct $f_0$. We focus on deep ReLU networks, motivated by the following example of Mhaskar et al. (2017, remark 8).

## 2.1 Motivating Example

Mhaskar et al. (2017, remark 8) shows that the bivariate function $f_{10}(x_1, x_2) = (x_1^2 x_2^2 - x_1^2 x_2 + 1)^{2^{10}}$ can be approximated more efficiently by a deep ReLU neural net than a shallow combination of ridge functions. To verify this observation, we simulate data from the following polynomial

$$f_1(x_1, x_2) = (x_1^2 x_2^2 - x_1^2 x_2 + 1)^2$$

where $(x_1, x_2)$ take values in $[-1, 1]^2$. We discretize the grid for a total training data of $201 \times 201 = 40401$ observations. There exists an exact Kolmogorov representation for this function as a superposition of semi-affine functions if we use the identities for the inner polynomial functions

$$x_1^2 x_2 = \frac{1}{2}(x_1^2 + x_2)^2 - \frac{1}{2}(x_1^2 - x_2)^2 \qquad (3)$$

$$(x_1 x_2)^2 = \frac{1}{4}(x_1 + x_2)^4 + \frac{7}{4 \cdot 3^3}(x_1 - x_2)^4 - \frac{1}{2 \cdot 3^3}(x_1 + 2x_2)^4 - \frac{2^3}{3^3}(x_1 + 2x_2)^4. \qquad (4)$$

Following the theoretical results of Mhaskar et al. (2017), we build an 11-layer deep ReLU network is used to approximate this polynomial. There are 9 units in the first hidden layer and 3 units in the further layers. All activation functions are ReLU. For comparison, we also build a shallow network with only 1 hidden layer but 2048 units. The MSE for the models, both trained with SGD in `TensorFlow` and `Keras`, are: 11 layers, 39 units with $MSE(train) = 0.0229, MSE(validation) = 0.0112$ and 1 layer, 2048 units with $MSE(train) = 0.0441, MSE(validation) = 0.09$. Both models outperform random forests.

## 3 Deep ReLU Networks

We now formally describe the generative model that gives rise to deep rectified linear unit networks. To fix notation, we write a deep neural network $f_{\boldsymbol{B}}^{DL}(\boldsymbol{x})$ as an iterative mapping specified by hierarchical layers of abstraction. With $L \in \mathbb{N}$ we denote the number of hidden layers and with $p_l \in \mathbb{N}$ the number of neurons at the $l^{th}$ layer. Setting $p_0 = p$ and $p_{L+1} = 1$, we denote with $\boldsymbol{p} = (p_0, \ldots, p_{L+1})' \in \mathbb{N}^{L+2}$ the vector of neuron counts for the entire network. The deep network is then characterized by a set of model parameters

$$\boldsymbol{B} = \{(\boldsymbol{W}_1, \boldsymbol{b}_1), (\boldsymbol{W}_2, \boldsymbol{b}_2), \ldots, (\boldsymbol{W}_L, \boldsymbol{b}_L)\}, \tag{5}$$

where $\boldsymbol{b}_l \in \mathbb{R}^{p_l}$ are shift vectors and $\boldsymbol{W}_L$ are $p_l \times p_{l-1}$ weight matrixes that link neurons between the $(l-1)^{th}$ and $l^{th}$ layers. Nodes in the ReLU network are connected through the following activation function $\sigma_{\boldsymbol{b}} : \mathbb{R}^r \to \mathbb{R}^r$

$$\sigma_{\boldsymbol{b}} \begin{pmatrix} y_1 \\ y_2 \\ \vdots \\ y_r \end{pmatrix} = \begin{pmatrix} \sigma(y_1 - b_1) \\ \sigma(y_2 - b_2) \\ \vdots \\ \sigma(y_r - b_r) \end{pmatrix},$$

where $\sigma(x) = ReLU(x) = \max(x, 0)$ denotes the rectified linear unit activation function.

Deep ReLU neural networks with $L$ layers and a vector of $\boldsymbol{p}$ hidden nodes define an input-output map $f_{\boldsymbol{B}}^{DL}(\boldsymbol{x}) : \mathbb{R}^p \to \mathbb{R}$ of the form

$$f_{\boldsymbol{B}}^{DL}(\boldsymbol{x}) = \boldsymbol{W}_{L+1} \sigma_{\boldsymbol{b}_L} \left( \boldsymbol{W}_L \sigma_{\boldsymbol{b}_{L-1}} \ldots \sigma_{\boldsymbol{b}_1}(\boldsymbol{W}_1 \boldsymbol{x}) \right). \tag{6}$$

The representation (6) casts neural networks as nested embeddings that allow to express the data flow through a network using variable-size data structures. Varying the number of active neurons allows a model to control the effective dimensionality for a given input and achieve desired approximation accuracy. Similarly as Schmidt-Hieber (2017), we will focus on a specific type of networks with an equal number of hidden neurons, i.e. $p_l = 12pN$ for each $1 \leq l \leq L$ for some $N \in \mathbb{N}$. We will see later in Section 5, that the optimal network width multiplier $N$ should relate to the dimensionality $p$ and smoothness $\alpha$.

## 4 Spike-and-Slab Regularization

We focus on uniformly bounded *s-sparse* deep nets with bounded parameters

$$\mathcal{F}(L, \boldsymbol{p}, s) = \left\{ f_{\boldsymbol{B}}^{DL}(\boldsymbol{x}) \text{ as in (6)} : \|f_{\boldsymbol{B}}^{DL}\|_\infty < F \text{ and } \|\boldsymbol{B}\|_\infty \leq 1 \text{ and } \|\boldsymbol{B}\|_0 \leq s \right\},$$

where $s \in \mathbb{N}$ is the sparsity level, i.e. an upper bound on the number of edges in the network, and where $F > 0$.

The amount of regularization needed to achieve optimal performance typically depends on unknown properties of functions one wishes to approximate such as their smoothness, compositional pattern and/or the number of variables they depend on. Hierarchical Bayes procedures have the potential to become fully adaptive and achieve (nearly) minimax performance, as if one knew these properties ahead of time. We will leverage the fully Bayes framework and devise a hierarchical procedure which can learn the optimal level of sparsity needed to achieve near-minimax rates of posterior convergence of neural networks. The cornerstone of this development will be the spike-and-slab framework.

Denote with

$$T = \sum_{l=0}^{L} p_{l+1}(p_l + 1) - p_{L+1} < (12 \, p \, N + 1)^{L+1} \tag{7}$$

the number of parameters in a fully connected network with $L$ layers and a vector of $\boldsymbol{p}$ neurons, where the inequality in (7) holds when $L \geq 2$ and $12pN \geq 2$. We treat the stacked vector of model coefficients $\boldsymbol{B} = (\beta_1, \ldots, \beta_T)'$ in (5) as a random vector arising from the *spike-and-slab* prior defined hierarchically through

$$\pi(\beta_j \mid \gamma_j) = \gamma_j \widetilde{\pi}(\beta_j) + (1 - \gamma_j)\delta_0(\beta_j), \tag{8}$$

where

$$\widetilde{\pi}(\beta) = \frac{1}{2}\mathbb{I}_{[-1,1]}(\beta)$$

is a uniform prior on an interval $[-1, 1]$, $\delta_0(\beta)$ is a dirac spike at zero, and where $\gamma_j \in \{0, 1\}$ for whether or not $\beta_j$ is nonzero. We collate the binary indicators into a vector $\boldsymbol{\gamma} = (\gamma_1, \ldots, \gamma_T)' \in \{0, 1\}^T$ that encodes the connectivity pattern. We assume that, given the sparsity level $s = |\boldsymbol{\gamma}|$, all architectures are equally likely a-priori, i.e.

$$\pi(\boldsymbol{\gamma} \mid s) = \frac{1}{\binom{T}{s}}. \tag{9}$$

The sparsity level $s$ will be first treated as fixed and later assigned a prior with exponential decay. The spike-and-slab construction (8) and (9) has been studied in linear models by Castillo and van der Vaart (2012) and in trees/forests by Rockova and van der Pas (2017), who showed that with a suitable prior on $s$, the posterior can adapt to the unknown level of sparsity. We conclude a very similar property for our proposed *spike-and-slab deep learning*.

It is worthwhile to point out that the prior (8) effectively zeroes out individual links rather than entire groups of links attached to one node. The second approach was explored by Ghosh and Doshi-Velez (2017), who suggested assigning a Horseshoe prior on the node preactivators, diminishing influence of individual neurons. The dropout procedure is also motivated as erasing nodes rather than links.

## 5 Posterior Concentration for Deep Learning

Reconstruction of $f_0$ from the training data $(Y_i, \boldsymbol{x}_i)_{i=1}^n$ can be achieved using a Bayesian approach. This requires placing a prior measure $\Pi(\cdot)$ on $\mathcal{F}(L, \boldsymbol{p}, s)$, the set of qualitative guesses of $f_0$. Given observed data $\boldsymbol{Y}^{(n)} = (Y_1, \ldots, Y_n)'$, inference about $f_0$ is then carried out via the posterior distribution

$$\Pi\left(A \mid \boldsymbol{Y}^{(n)}, \{\boldsymbol{x}_i\}_{i=1}^n\right) = \frac{\int_A \prod_{i=1}^n \Pi_f(Y_i \mid \boldsymbol{x}_i)\mathrm{d}\,\Pi(f)}{\int \prod_{i=1}^n \Pi_f(Y_i \mid \boldsymbol{x}_i)\mathrm{d}\,\Pi(f)} \quad \forall A \in \mathcal{B},$$

where $\mathcal{B}$ is a $\sigma$-field on $\mathcal{F}(L, \boldsymbol{p}, s)$ and where $\Pi_f(Y_i \mid \boldsymbol{x}_i)$ is the likelihood function for the output $Y_i$ under $f$.

Our goal is to determine *how fast the posterior probability measure concentrates around $f_0$* as $n \to \infty$. This speed can be assessed by inspecting the size of the smallest $\|\cdot\|_n$-neighborhoods around $f_0$ that contain most of the posterior probability (Ghosal and van der Vaart, 2007). For a diameter $\varepsilon > 0$ and some $M > 0$, we denote with

$$A_{\varepsilon,M} = \{f_{\boldsymbol{B}}^{DL} \in \mathcal{F}(L, \boldsymbol{p}, s) : \|f_{\boldsymbol{B}}^{DL} - f_0\|_n \leq M\,\varepsilon\}$$

the $M\varepsilon$-neighborhood centered around $f_0$. Our goal is to show that

$$\Pi(A_{\varepsilon_n,M_n}^c \mid \boldsymbol{Y}^{(n)}) \to 0 \quad \text{in } \mathbb{P}_{f_0}^{(n)}\text{-probability as } n \to \infty \tag{10}$$

for any $M_n \to \infty$ and for $\varepsilon_n \to 0$ such that $n\,\varepsilon_n^2 \to \infty$. We will position our results using $\varepsilon_n = n^{-\alpha/(2\alpha+p)}\log^\delta(n)$ for some $\delta > 0$, the near-minimax rate for a $p$-dimensional $\alpha$-smooth function. Proving techniques for statements of type (10) were established in several pioneering works including Ghosal, Ghosh and van der Vaart (2000), Ghosal and van der Vaart (2007), Shen and Wassermann (2001), Wong and Shen (1995), Walker et al. (2007).

The statement (10) can be proved by verifying the following three conditions (suitably adapted from Theorem 4 of Ghosal and van der Vaart (2007)):

$$\sup_{\varepsilon > \varepsilon_n} \log \mathcal{E}\left(\tfrac{\varepsilon}{36}; A_{\varepsilon,1} \cap \mathcal{F}_n; \|\cdot\|_n\right) \leq n\,\varepsilon_n^2 \tag{11}$$

$$\Pi(A_{\varepsilon_n,1}) \geq e^{-d\,n\,\varepsilon_n^2} \tag{12}$$

$$\Pi(\mathcal{F}\backslash\mathcal{F}_n) = o(e^{-(d+2)\,n\,\varepsilon_n^2}) \tag{13}$$

for some $d > 2$. Above, $\mathcal{F}_n \subseteq \mathcal{F}(L, \boldsymbol{p}, s)$ is an approximating space (sieve) that captures the essence of the parameter space. Condition (11) restricts the size of the model as measured by the Le Cam dimension (or local entropy). The Le Cam dimension, defined here in terms of the log-covering number of $A_{\varepsilon,1} \cap \mathcal{F}_n$, gives rise to the minimax rate of convergence under certain conditions (Le Cam, 1973). The sieve should not be too large (Condition (11)), it should be rich enough to approximate $f_0$ well and it should receive most of the prior mass (Condition (13)).

The prior concentration Condition (12) is needed to make sure that the prior rewards shrinking neighborhoods of $f_0$. This requirement is a bit at odds with Condition (11). The richer the model class (i.e. the more layers/neurons), the better the approximation to $f_0$. It is essential that the prior is supported on models that are good approximators, but that do not overfit. It is commonly agreed (Ghosal and van der Vaart, 2007) that the approximation gap should be no larger than a constant multiple of $\varepsilon_n$. Below, we review some known results about expressibility of neural networks to get insights into how many layers/neurons are needed to achieve the desired level of approximation accuracy.

## 5.1 Function Class Approximation Rates

There is an extensive literature on the approximation properties of neural nets. Many tight approximation results are available for simple functions such as indicators $f(\boldsymbol{x}) = \mathbb{I}_B(\boldsymbol{x})$ where $B$ is a unit ball (Cheang and Barron, 2000) or a half-space (Cheang (2010), Kainen et al. (2003, 2007) and Křkova et al. (1997)). Recent results on the efficiency of ridge NNs (which arise as shallow learners of the form $f = \sum_{j=1}^n a_j \sigma(w_j^T x - b_j)$ for sigmoidal $\sigma(\cdot)$) are available in Ismailov(2017), Klusowski and Barron (2016, 2017). Pinkus (1999) and Petrushev (1999) provide some of the early bounds.

In general, one tries to characterize the asymptotic behavior of the approximation error as follows:

$$\|f_0 - \widehat{f}\| = \mathcal{O}(N^{-\frac{\alpha}{p}}) \iff \|f_0 - \widehat{f}\| \leq \varepsilon \text{ where } N = \mathcal{O}(\varepsilon^{-\frac{p}{\alpha}}), \tag{14}$$

where $f_0$ is a real-valued $\alpha$-smooth function, $\widehat{f}$ is the neural-network reconstruction and where $N$ is the "size" of the network (typically the number of hidden nodes). Different bounds can be obtained for different classes of $f_0$ and different norms $\|\cdot\|$. The goal is to assess how complex the network ought to be for it to approximate $f_0$ well (up to a constant multiple of $\varepsilon_n$).

For deep networks, one also wants to find the asymptotic behavior of the approximation error as a function of depth, not only its size. The following Lemma will be an essential building block in the proof of our main theorem. It summarizes the expressibility of deep ReLU networks by linking their approximation error (when estimating Hölder smooth functions) to the network depth, width and sparsity.

**Lemma 5.1.** *(Schmidt-Hieber, 2017) Assume that $f_0 \in \mathcal{H}_p^\alpha$ for some $\alpha > 0$. Then for any $N \geq (\alpha+1)^p \vee (\|f_0\|_{\mathcal{H}}^\alpha + 1)$ there exists a neural network $\widehat{f} \in \mathcal{F}(L^\star, \boldsymbol{p}_N^{L^\star} = (p, 12pN, \ldots, 12pN, 1), s^\star)'$ with*

$$L^\star = 8 + (\lfloor \log_2(n) \rfloor + 5)(1 + \lceil \log_2 p \rceil) \tag{15}$$

*layers and sparsity level $s^\star$ satisfying*

$$s^\star \leq 94\,p^2(\alpha+1)^{2p}N\,(L^\star + \lceil \log_2 p \rceil) \tag{16}$$

*such that*

$$\|\widehat{f} - f_0\|_\infty \leq (2\|f_0\|_{\mathcal{H}^\alpha} + 1)3^{p+1}\frac{N}{n} + \|f_0\|_{\mathcal{H}^\alpha}2^\alpha N^{-\alpha/p}.$$

*Proof.* Apply Theorem 3 of Schmidt-Hieber (2017) with $m = \lfloor \log_2(n) \rfloor$.

**Remark 5.1.** *In a related result, Yarotsky (2017) shows that there exists a ReLU network that satisfies $\|f - \widehat{f}^{DL}\|_\infty \leq \varepsilon$ with sparsity $s = c \cdot \varepsilon^{-\frac{p}{\alpha}} / \log_2(1/\varepsilon) + 1$ and depth $L = c \cdot (\log_2(1/\varepsilon) + 1)$ where $c = c(p, \alpha)$. Petersen and Voigtlaender (2017) extend this result to $L^2$-smooth functions.*

We assume that $p = O(1)$ as $n \to \infty$. Lemma (5.1) essentially states that in order to approximate an $\alpha$-Hölder smooth function with an error that is at most a constant multiple of $\varepsilon_n$, we have to choose $L \asymp \log(n)$ layers with sparsity $s \leq C_S \lfloor n^{p/(2\alpha+p)} \rfloor$. This follows by setting $N = C_N \lfloor n^{p/(2\alpha+p)} / \log(n) \rfloor$.

# 6 Posterior Concentration for Sparse ReLU Networks

We formalize large sample statistical properties of posterior distributions over ReLU networks. First, we consider a hierarchical prior distribution on $\mathcal{F}(L, \boldsymbol{p}, s)$, keeping $L$, $\boldsymbol{p}$ and $s$ fixed as if they were known. The prior distribution now only consists of the prior on the connectivity pattern (9) and the spike-and-slab prior on the weights/offsets (8).

Our first result provides guidance for calibrating Bayesian deep sparse ReLU networks (choosing the sparsity level and the number of neurons) *when the level of smoothness $\alpha$ is known.* The result can be regarded as a Bayesian analogue of Theorem 1 of Schmidt-Hieber (2017), who showed near-minimax rate-optimality of a sparse multilayer ReLU network estimator that minimizes empirical least-squares. This was the first result on rate-optimality of deep ReLU networks in non-parametric regression, obtained assuming that the sparsity $s$ is known and that the function $f_0$ is a composition of Hölder functions. We build on this result and show that the *entire posterior distribution* for deep sparse ReLU neural networks is concentrating at the near-minimax rate, when $\alpha$ is known and when $f_0$ is a Hölder smooth function. In the next section, we provide an adaptive result which no longer requires the knowledge of $\alpha$.

**Theorem 6.1.** *(Deep ReLUs are near-minimax.) Assume $f_0 \in \mathcal{H}_p^\alpha$, where $p = O(1)$ as $n \to \infty$, $\alpha < p$ and $\|f_0\|_\infty \leq F$. Let $L^\star$ be as in (15), $s^\star$ as in (16) and $\boldsymbol{p}^\star = (p, 12pN^\star, \ldots, 12pN^\star, 1)' \in \mathbb{N}^{L^\star+2}$, where $N^\star = C_N \lfloor n^{p/(2\alpha+p)} / \log(n) \rfloor$. Then the posterior probability concentrates at the rate $\varepsilon_n = n^{-\alpha/(2\alpha+p)} \log^\delta(n)$ for $\delta > 1$ in the sense that*

$$\Pi(f_{\boldsymbol{B}}^{DL} \in \mathcal{F}(L^\star, \boldsymbol{p}^\star, s^\star) : \|f - f_0\|_n > M_n \varepsilon_n \mid \boldsymbol{Y}^{(n)}) \to 0 \tag{17}$$

*in $\mathbb{P}_0^n$ probability as $n \to \infty$ for any $M_n \to \infty$.*

*Proof.* Supplementary Materials □

**Remark 6.1.** *In Theorem 5.1, we do not need to construct a sieve $\mathcal{F}_n$, because $s$ and $N$ are fixed. We can simply take $\mathcal{F}_n = \mathcal{F}(L^\star, p^\star, s^\star)$ in which case $\mathcal{F} \backslash \mathcal{F}_n = \emptyset$ and (13) holds trivially.*

Theorem 6.1 continues the line of theoretical investigation of Bayesian machine learning procedures. Lee (2000) obtained posterior consistency for single-layer sigmoidal networks. van der Pas and Rockova (2017) and Rockova and van der Pas (2017) obtained concentration results for Bayesian regression trees and forests. Compared to these developments, deep neural networks (NN) seem to be more flexible in estimating smooth regression functions. Indeed, trees or forests are ultimately step function approximators and, as such, are near-minimax only for $0 < \alpha \leq 1$ (Rockova and van der Pas, 2017). As we have shown in Theorem 6.1, Bayesian deep ReLU networks are near-minimax when $0 < \alpha < p$, where $p$ can be much larger than 1. One practical implication is that one would expect NN's to outperform trees for very smooth objects.

## 6.1 Bayesian Deep Learning Adapts to Smoothness

Theorem 6.1 was conceived for network architectures that are optimally tuned for $\alpha$ that is fixed as if it were known. However, such oracle information is rarely available, rendering the result less relevant for practical design of networks. In this section, we devise a hierarchical prior construction (by endowing the unknown network parameters with suitable priors), under which the posterior performs *as well as if we knew $\alpha$.*

From the previous section (and discussion in Schmidt-Hieber (2017)), we know that the number of layers $L$ can be chosen without the knowledge of smoothness $\alpha$. We will thus continue to assume that the number of layers is fixed and equal to $L^\star$ in (15).

Both the network width $N$ and sparsity level $s$ were chosen in an $\alpha$-dependent way. To obviate this constraint, we treat them as unknown with the following priors. For the network width multiplier $N$, we deploy

$$\pi(N) = \frac{\lambda^N}{(\mathrm{e}^\lambda - 1)N!} \quad \text{for} \quad N = 1, 2, \ldots \quad \text{for some} \quad \lambda \in \mathbb{R}. \tag{18}$$

The prior (18) is one of the classical complexity priors used frequently in the Bayesian non-parametric literature (Coram and Lalley (2006), Liu et al. (2017), Rockova and van der Pas (2017)). Similarly, the sparsity level $s$ will be now treated as unknown with the following prior

$$\pi(s) \propto e^{-\lambda_s s} \quad \text{for} \quad \lambda_s > 0. \tag{19}$$

Denote with $\boldsymbol{p}_N^{L^\star} = (p, 12pN, \dots, 12pN, 1)' \in \mathbb{N}^{L^\star}$ the now random vector of network widths that depend on $N$ and $L^\star$. Our parameter space now consists of shells of *sparse* deep nets with different widths and sparsity levels, i.e.

$$\mathcal{F}(L^\star) = \bigcup_{N=1}^{\infty} \bigcup_{s=0}^{T} \mathcal{F}(L^\star, \boldsymbol{p}_N^{L^\star}, s),$$

where $T$ is the number of links in a fully connected network (defined in (7)). We will design an approximating sieve as follows:

$$\mathcal{F}_n = \bigcup_{N=1}^{N_n} \bigcup_{s=0}^{s_n} \mathcal{F}(L^\star, \boldsymbol{p}_N^{L^\star}, s) \tag{20}$$

for some suitable $N_n \in \mathbb{N}$ and $s_n \leq T$. Following our discussion earlier in this section, the sieve $\mathcal{F}_n$ should be rich enough to include networks that approximate well. To this end, we choose $N_n$ and $s_n$ similar to the "optimal choices" obtained from the fixed $\alpha$ case, i.e.

$$N_n = \lfloor \widetilde{C}_N n^{p/(2\alpha+p)} \log^{2\delta-1}(n) \rfloor \asymp n\varepsilon_n^2/\log n, \quad \text{and} \quad s_n = \lfloor L^\star N_n \rfloor \asymp n\varepsilon_n^2$$

for $\widetilde{C}_N > 0$. With these choices, we show that the posterior distribution concentrates at the same rate as before, but without assuming $\alpha$.

**Theorem 6.2.** *(Deep ReLUs adapt to smoothness.) Assume $f_0 \in \mathcal{H}_p^\alpha$, where $p = O(1)$ as $n \to \infty$, $\alpha < p$, and $\|f_0\|_\infty \leq F$. Let $L^\star$ be as in (15) and assume priors (19) and (18). Then the posterior probability concentrates at the rate $\varepsilon_n = n^{-\alpha/(2\alpha+p)} \log^\delta(n)$ for $\delta > 1$ in the sense that*

$$\Pi(f_{\boldsymbol{B}}^{DL} \in \mathcal{F}(L^\star) : \|f_{\boldsymbol{B}}^{DL} - f_0\|_n > M_n \varepsilon_n \,|\, \boldsymbol{Y}^{(n)}) \to 0 \tag{21}$$

*in $\mathbb{P}_0^n$ probability as $n \to \infty$ for any $M_n \to \infty$.*

*Proof.* Supplementary Materials.

Theorem 6.2 has a very important implication. It shows that, once we assign suitable complexity priors over the network size and sparsity, we can perform as well as if we knew the smoothness $\alpha$. This type of adaptation for deep learning is, to the best of our knowledge, a new phenomenon. It originates from the fully Bayesian treatment of deep learning. Similar adaptations were obtained for Bayesian forests (Rockova and van der Pas (2017)), where the adaptation costs only a small fraction of the log factor. Here, we have the same rate as in the non-adaptive case, suggesting that the analysis could be potentially refined a bit to obtain a sharper rate when $\alpha$ is known.

We conclude the paper with the following important corollary stating that Bayesian deep ReLU networks with adaptive spike-and-slab priors *do not overfit* in the sense that the posterior probability of using more than the optimal number of nodes and links goes to zero as $n \to \infty$

**Corollary 6.1.** *(Deep ReLUs do not overfit.) Under the assumptions in Theorem 6.2 we have*

$$\Pi(N > N_n \,|\, \boldsymbol{Y}^{(n)}) \to 0 \quad \text{and} \quad \Pi(s > s_n \,|\, \boldsymbol{Y}^{(n)}) \to 0 \tag{22}$$

*in $\mathbb{P}_0^n$ probability as $n \to \infty$.*

*Proof.* This statement follows from Lemma 1 of Ghosal and van der Vaart (2007) and holds upon the satisfaction of the conditions

$$\Pi(N > N_n) = o(e^{-(d+2)n\varepsilon_n^2}) \quad \text{and} \quad \Pi(s > s_n) = o(e^{-(d+2)n\varepsilon_n^2})$$

that are verified in Supplementary Materials.

The key observation behind Corollary 5.1 is that the posterior *does not overshoot* in terms of the width and sparsity, rewarding only *small* networks that are *sparse*. That is, the posterior concentrates on networks with up to the optimal number $s_n$ of links. This is purely a by-product of Bayesian regularization and, again, this property does not rely on any oracle information about $\alpha$.

### 6.2   Implementation Considerations

For a fixed architecture (i.e. $N$ is non-random) and continuous spike-and-slab priors, one could perform an Expectation-Maximization algorithm by iteratively (a) deploying SGD with $\ell_1/\ell_2$ regularization and coefficient specific penalties (M-step) and (b) computing conditional probability that the coefficient is non-negligible (E-step). The E-step is inexpensive and determines, one coefficient at a time, how much shrinkage should be deployed. The M-step can be readily obtained with existing software. Such an EM strategy has been successfuly deployed in linear models (Rockova and George (2014, 2018) and related strategies have already been deployed for neural networks (via Variational Bayes by Ullrich et al. (2017) or with *Bayes by Backprop* by Blundell et al. (2015)). The optimization strategy is feasible for Gaussian/Laplace mixtures which are continuous approximations of the the point-mass mixture prior that we analyze. Turning optimization into posterior sampling is feasible with a weighted Bayesian bootstrap (Newton, Polson and Xu (2018)). Attaching a random weight to each observation in the likelihood, modes of resulting posteriors constitute samples from the original (unweighted) posterior. Regarding the adaptive architectures (when $N$ is random), they *can* be learned as well using ideas from Liu, Rockova and Wang (2018).

## 7   Closing Remarks

The goal of this paper was to study posterior concentration for Bayesian deep learning and to provide new theoretical justifications for neural networks from a Bayesian point of view. Our theoretical results can be summarized in three points. First, in Theorem 6.1 we show that Bayesian deep ReLU networks can be near-minimax, if tuned properly. Second, in Theorem 6.2 we show that, by assigning suitable complexity priors over the network architecture, Bayesian deep ReLU networks can be near-minimax tuning-free. In other words, they can adapt to unknown smoothness, giving rise posteriors that concentrate around smooth surfaces at near-minimax rates. Third, in Corollary 6.1 we provide some arguments for why Bayesian deep ReLU networks are less eager to overfit. The key ingredients for these results were (a) sparsity through spike-and-slab regularization, (b) complexity priors on the network width and sparsity level. Posterior concentration rate results of this type are now slowly entering the machine learning community as a tool for (a) obtaining more insights into Bayesian methods (van der Pas and Rockova (2017), Rockova and van der Pas (2017)) and (b) prior calibrations.

There are many non-parametric methods that can achieve near-minimax recovery of Hölder smooth functions. The appeal of deep learning is their compositional structure which makes them ideal for regression surfaces that are themselves compositions. Indeed, there is evidence that deep learning has exponential advantage over shallow networks for approximating compositions. Schmidt-Hieber (2017) showed that sparsely connected deep ReLU networks achieve a near-minimax rate in learning for compositions of smooth functions. It is possible to adapt our techniques to obtain a Bayesian analogue of his compositional result.

Generalizing the results to other activators is possible, provided that one can show that Hölder smooth maps can be approximated well with sufficiently small networks. One could follow the general recipe from Section 5 for e.g. sigmoidal functions. We focused on ReLU since they are typically preferred over sigmoidal.

## 8   Acknowledgments

This work was supported by the James S. Kemper Research Fund at the University of Chicago Booth School of Business. The authors would like to thank the anonymous referees and the area chair for useful feedback.

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
