[Supplementary Material]

# Posterior Concentration for Sparse Deep Learning

**Nicholas G. Polson and Veronika Ročková**
Booth School of Business
University of Chicago
Chicago, IL 60637

## 1    Supplemental Materials

### 1.1    Proof of Theorem 6.1

We prove the theorem by verifying Condition (11) and (12), setting $\mathcal{F}_n = \mathcal{F}(L^\star, \boldsymbol{p}^\star, s^\star)$. First, we need to verify the entropy condition and show that

$$\sup_{\varepsilon > \varepsilon_n} \log \mathcal{E}\left(\tfrac{\varepsilon}{36}, \{f_{\boldsymbol{B}}^{DL} \in \mathcal{F}(L^\star, \boldsymbol{p}^\star, s^\star) : \|f - f_0\|_n < \varepsilon\}, \|.\|_n\right) \leq n\,\varepsilon_n^2. \tag{1}$$

We can upper-bound the local entropy (1) with the global metric entropy. In addition, because

$$\{f_{\boldsymbol{B}}^{DL} \in \mathcal{F}(L^\star, \boldsymbol{p}^\star, s^\star) : \|f\|_\infty \leq \varepsilon\} \subset \{f_{\boldsymbol{B}}^{DL} \in \mathcal{F}(L^\star, \boldsymbol{p}^\star, s^\star) : \|f\|_n \leq \varepsilon\},$$

we can upper-bound (1) with

$$\log \mathcal{E}\left(\tfrac{\varepsilon_n}{36}, f_{\boldsymbol{B}}^{DL} \in \mathcal{F}(L^\star, \boldsymbol{p}^\star, s^\star), \|.\|_\infty\right) \leq (s^\star + 1) \log\left(\frac{72}{\varepsilon_n}(L^\star + 1)(12pN + 1)^{2(L^\star + 2)}\right)$$

$$\lesssim n^{p/(2\alpha + p)} \log(n) \log\left(n/\log^\delta(n)\right) \lesssim n^{p/(2\alpha + p)} \log^2(n) \lesssim n\varepsilon_n^2$$

for $\delta > 1$, where we used Lemma 10 of Schmidt-Hieber (2017) and the fact that $s^\star \lesssim n^{p/(2\alpha + p)}$ and $N \asymp n^{p/(2\alpha + p)}/\log(n)$. This verifies the entropy Condition (11).

Next, we want to show that the prior concentrates enough mass around the truth in the sense that

$$\Pi(f_{\boldsymbol{B}}^{DL} \in \mathcal{F}(L^\star, \boldsymbol{p}^\star, s^\star) : \|f_{\boldsymbol{B}}^{DL} - f_0\|_n \leq \varepsilon_n) \geq \mathrm{e}^{-d\,n\,\varepsilon_n^2} \tag{2}$$

for some $d > 2$. Choosing $N^\star = C_N \lfloor n^{p/(2\alpha + p)}/\log(n)\rfloor$ in Lemma 5.1, there exists a neural network $\widehat{f}_{\widehat{\boldsymbol{B}}} \in \mathcal{F}(L^\star, \boldsymbol{p}^\star, s^\star)$ consisting of $\boldsymbol{p}^\star$ nodes aligned in $L^\star \lesssim \log(n)$ layers and indexed by $\|\widehat{\boldsymbol{B}}\|_0 = s^\star \lesssim n^{p/(2\alpha + p)} \log(n)$ nonzero parameters such that

$$\|\widehat{f}_{\widehat{\boldsymbol{B}}} - f_0\|_n \leq C_\infty n^{-\alpha/(2\alpha + p)} \log^{\delta\alpha/p}(n) \lesssim \varepsilon_n/2,$$

where the last inequality follows from $\alpha < p$, absorbing $C_\infty$ in the concentration rate. The approximation $\widehat{f}_{\widehat{\boldsymbol{B}}}$ sits on a network architecture characterized by a specific pattern $\widehat{\gamma}$ of nonzero links among $\boldsymbol{B}$, i.e. $\widehat{W}_l$ and $\widehat{a}_l$ for $1 \leq l \leq L + 1$. We denote by $\mathcal{F}(\widehat{\gamma}, L^\star, \boldsymbol{p}^\star, s^\star) \subset \mathcal{F}(L^\star, \boldsymbol{p}^\star, s^\star)$ all the functions supported on this particular architecture. These functions differ only in the size of the $s^\star$ nonzero coefficients among $\boldsymbol{B}$, denoted by $\boldsymbol{\beta} \in \mathbb{R}^{s^\star}$. With $\widehat{\boldsymbol{\beta}}$, we denote the $s^\star$-vector associated with the nonzero elements in $\widehat{\boldsymbol{B}}$.

Note that there are $\binom{T}{s^\star} \leq (12\,p\,N)^{(L^\star + 1)\,s^\star}$ combinations to pick $s^\star$ the nonzero coefficients and each one, according to prior (9), has an equal prior probability of occurrence $\frac{1}{\binom{T}{s^\star}}$.

To continue, we note (from the triangle inequality) that

$$\{f_{\boldsymbol{B}}^{DL} \in \mathcal{F}(L^\star, \boldsymbol{p}^\star, s^\star) : \|f_{\boldsymbol{B}}^{DL} - f_0\|_n \leq \varepsilon_n\} \supset \{f_{\boldsymbol{B}}^{DL} \in \mathcal{F}(\widehat{\gamma}) : \|f_{\boldsymbol{B}}^{DL} - \widehat{f}_{\widehat{\boldsymbol{B}}}\|_\infty \leq \varepsilon_n/2\}.$$

Next, we denote with $\{\boldsymbol{\beta} \in \mathbb{R}^{s^{\star}} : \|\boldsymbol{\beta}\|_{\infty} \leq 1 \quad \text{and} \quad \|\boldsymbol{\beta} - \widehat{\boldsymbol{\beta}}\|_{\infty} \leq \varepsilon_n\}$ the set of coefficients that are at most $\varepsilon$-away from the best approximating coefficients $\widehat{\boldsymbol{\beta}}$ of the neural network $\widehat{f}_{\widehat{\boldsymbol{B}}} \in \mathcal{F}(\widehat{\boldsymbol{\gamma}}, L^{\star}, \boldsymbol{p}^{\star}, s^{\star})$. From the proof of Lemma 10 of Schmidt-Hieber (2017), it follows that

$$\left\{ f_{\boldsymbol{B}}^{DL} \in \mathcal{F}(\widehat{\boldsymbol{\gamma}}) : \|f_{\boldsymbol{B}}^{DL} - \widehat{f}_{\widehat{\boldsymbol{B}}}\|_{\infty} \leq \frac{\varepsilon_n}{2} \right\} \supset$$

$$\left\{ \boldsymbol{\beta} \in \mathbb{R}^{s^{\star}} : \|\boldsymbol{\beta}\|_{\infty} \leq 1 \text{ and } \|\boldsymbol{\beta} - \widehat{\boldsymbol{\beta}}\|_{\infty} \leq \frac{\varepsilon_n}{2V(L^{\star} + 1)} \right\},$$

where $V = \prod_{l=0}^{L^{\star}+1}(p_l^{\star} + 1)$. Now we have all the pieces needed to find a lower bound to the probability in (2). We can write, for some suitably large $C > 0$,

$$\Pi\left(f_{\boldsymbol{B}}^{DL} \in \mathcal{F}(L^{\star}, \boldsymbol{p}^{\star}, s^{\star}) : \|f_{\boldsymbol{B}}^{DL} - f_0\|_n \leq \varepsilon_n\right) > \frac{\Pi(f_{\boldsymbol{B}}^{DL} \in \mathcal{F}(\widehat{\boldsymbol{\gamma}}, L^{\star}, \boldsymbol{p}^{\star}, s^{\star}) : \|f_{\boldsymbol{B}} - \widehat{f}_{\widehat{\boldsymbol{B}}}\|_{\infty} \leq \varepsilon_n/2)}{\binom{T}{s^{\star}}}$$

$$> e^{-(L^{\star}+1)s^{\star}\log(12\,p\,N^{\star})}\Pi\left(\boldsymbol{\beta} \in \mathbb{R}^{s^{\star}} : \|\boldsymbol{\beta}\|_{\infty} \leq 1 \text{ and } \|\boldsymbol{\beta} - \widehat{\boldsymbol{\beta}}\|_{\infty} \leq \frac{\varepsilon_n}{2V(L^{\star} + 1)}\right).$$

To continue to lower-bound the expression above, we note that

$$e^{-(L^{\star}+1)s^{\star}\log(12\,p\,N^{\star})} > e^{-C\log^2(n)n^{p/(2\alpha+p)}}$$

for some $C > 0$. Under the uniform prior distribution on a cube $[-1, 1]^{s^{\star}}$ we can write

$$\Pi\left(\boldsymbol{\beta} \in \mathbb{R}^{s^{\star}} : \|\boldsymbol{\beta}\|_{\infty} \leq 1 \text{ and } \|\boldsymbol{\beta} - \widehat{\boldsymbol{\beta}}\|_{\infty} \leq \frac{\varepsilon_n}{2V(L^{\star} + 1)}\right) = \left(\frac{\varepsilon_n}{2V(L^{\star} + 1)}\right)^{s^{\star}}$$

$$\geq e^{-s^{\star}(L^{\star}+2)\log(12\,p\,n/\log^{\delta}(n))} \geq e^{-D\,n^{p/(2\alpha+p)}\log^2(n)}$$

for some $D > 0$. We can combine this bound with the preceding expressions to conclude that $e^{-(C+D)\,n^{p/(2\alpha+p)}\log^2(n)} \geq e^{-d\,n\,\varepsilon_n^2}$ for $\delta > 1$ and $d > C + D$. This concludes the proof of (17).

## 1.2   Proof of Theorem 6.2

First we show that the sieve $\mathcal{F}_n$ defined in (20) is still reasonably small in the sense that the log covering number can be upper-bounded by a constant multiple of $n^{p/(2\alpha+p)}\log^{2\delta}(n)$. It follows from the proof of Theorem 6.1 that the global metric entropy satisfies

$$\mathcal{E}\left(\tfrac{\varepsilon_n}{36}, \mathcal{F}_n, \|.\|_n\right) \leq \sum_{N=1}^{N_n}\sum_{s=0}^{s_n} e^{(s+1)\log\left(\frac{72}{\varepsilon_n}(L^{\star}+2)(12pN+1)^{2(L^{\star}+2)}\right)}$$

$$\lesssim N_n\,s_n\,e^{C\,(L^{\star}+1)(s_n+1)\log(pN_nL^{\star}/\varepsilon_n)}$$

for some $C > 0$ and thereby

$$\log\mathcal{E}\left(\tfrac{\varepsilon_n}{36}, \mathcal{F}_n, \|.\|_n\right) \lesssim \log N_n + \log s_n + n\,\varepsilon_n^2 \lesssim n\,\varepsilon_n^2.$$

This verifies Condition (11).

Next, we need to show that the prior charges the sieve in the sense that $\Pi[\mathcal{F}_n^c] = o(e^{(d+2)n\varepsilon_n^2})$ for some $d > 2$ (determined below). We have

$$\Pi[\mathcal{F}_n^c] < \Pi(N > N_n) + \Pi(s > s_n).$$

We apply the Chernoff bound to find that

$$\Pi(N > N_n) < e^{-t\,(N_n+1)}\mathbb{E}\,e^{t\,N} \propto e^{-t\,(N_n+1)}\left(e^{e^t\lambda} - 1\right) \tag{3}$$

for any $t > 0$. With our choice $N_n = \lfloor\widetilde{C}_N n^{p/(2\alpha+p)}\log^{2\delta-1}n\rfloor$ and with $t = \log N_n$ we obtain

$$\Pi(N > N_n)e^{(d+2)\,n\varepsilon_n^2} \lesssim e^{-(N_n+1)\log N_n+\lambda N_n+(d+2)\,n\varepsilon_n^2} \to 0$$

for a large enough constant $\widetilde{C}_N$. Next, we find that

$$\Pi(s > s_n)e^{(d+2)\,n\varepsilon_n^2} \lesssim e^{-C_s(\lfloor L^{\star}N_n\rfloor+1)+(d+2)\,n\varepsilon_n^2} \to 0$$

for some suitably large $\widetilde{C}_N > 0$. This verifies Condition (13).

Finally, we verify the prior concentration Condition (12). For $N^\star < N_n$ and $s^\star < s_n$ we know from the proof of Theorem 6.1 that

$$\Pi(f_{\boldsymbol{B}}^{DL} \in \mathcal{F}(L^\star, \boldsymbol{p}^\star, s^\star) : \|f_{\boldsymbol{B}}^{DL} - f_0\|_n \leq \varepsilon_n) \geq \mathrm{e}^{-D_1 \, n \, \varepsilon_n^2}$$

for some $D_1 > 2$. Our priors put enough mass at the "right choices" $(N^\star, s^\star)$ in the sense that $\pi(N^\star) \gtrsim \mathrm{e}^{-N_n \log(N_n/\lambda)} \gtrsim \mathrm{e}^{-D \, n \varepsilon_n^2}$ and $\pi(s^\star) \gtrsim \mathrm{e}^{-D \, n \varepsilon_n^2}$ for some suitable $D > 0$. Then we can write

$$\Pi(f_{\boldsymbol{B}}^{DL} \in \mathcal{F}_n : \|f_{\boldsymbol{B}}^{DL} - f_0\|_n \leq \varepsilon_n)$$
$$\geq \pi(N^\star)\pi(s^\star)\Pi(f_{\boldsymbol{B}}^{DL} \in \mathcal{F}(L^\star, \boldsymbol{p}^\star, s^\star) : \|f_{\boldsymbol{B}}^{DL} - f_0\|_n \leq \varepsilon_n) \geq \mathrm{e}^{-(2D+D_1) \, n\varepsilon_n^2}.$$

With these considerations, we conclude the proof of Theorem 6.2.