[Reviews · NeurIPS 2018]

Reviewer 1



The paper presents few results on the concentration rates of the posterior distribution of the parameters of ReLU networks, along the line of the ones already available in a non-Bayesian setting. My overall score is due to the fact that, in my view, the organisation of the paper is not clear. The new results are jut mentioned at the end of the paper, with few comments, the prior distributions are chosen without cross-references to any known rule. The introduction present a description of the literature, without a presentation of the problem, which is given only in Section 2 (i.e. after the introduction). The authors would like to present the results as applicable to more deep learning techniques, but lack in showing why they are generalisable to cases different from ReLU networks. From the point of view of the clarity, there are a lot of acronyms and mathematical symbols which are never defined, making the paper difficult to read for someone who is not familiar with the subject: since the focus of the paper is between deep-learning and Bayesian nonparametrics the reader possibly interested in it may be not familiar with part of the literature/notation. This point is particularly important given my previous comment on the fact that the definitions are given in a way which is not linear.

Reviewer 2



Summary: ------------ This paper significantly extends known results and work regarding the consistency and rate-optimality of posteriors associated with neural networks, in particular focusing on contemporarily-relevant deep architectures with ReLUs as opposed to sigmoidal units. By imposing the classical spike-and-slab prior over models, a fully Bayesian analysis is enabled, whereby the model-averaged posterior is shown to converge to a neighborhood of the true regression function at a minimax-optimal rate. The analysis first focuses on deep ReLU networks with a uniform number of hidden units per layer where the sparsity, network parameters, and outputs are bounded. At a fixed setting of number of layers, number of units, and sparsity, the first key result is a statement that the posterior measure of lying outside of an \epsilon_n M_n neighborhood of f_0 vanishes in probability, where \epsilon_n is (~) the minimax rate for an \alpha-smooth function. The specific statement under the architectural and sparsity assumptions appears in Theorem 5.1, which is proved by verifying conditions (11-13) of Ghosal and van der Vaart, which is in turn done by invoking Lemma 5.1 (Schmidt-Hieber) showing the existence of a deep ReLU network approximating an \alpha-smooth function with bounded L_\infty norm. Unfortunately, this result prescribes setting the sparsity (s) and network size (N) as a function of \alpha, knowledge of which does not usually occur in practice. To circumvent this, a hierarchical Bayes procedure is devised by placing a Poisson prior on the network size and an exponential prior over the sparsity. Using an approximating sieve, posterior under this hierarchical prior attains the same rate as the case that the smoothness was known, and that furthermore, the posterior measure that both the optimal network size and sparsity are exceeded goes to zero in probability. This is a high-quality theoretical paper, which I believe also did a very nice job of practically motivating its theoretical contributions. Theoretical investigation into the behavior of deep networks is proceeding along many fronts, and this work provides valuable insights and extensions to the works mentioned above. Clarity: --------- The paper is very well-written and clear. My background is not in the approximation theory of neural networks, but with my working understanding of Bayesian decision theory I had few problems following the work and its line of reasoning. Significance: ------------ As mentioned, the paper is timely and significant. Technical Correctness: ---------------------- I did not painstakingly verify each detail of the proofs, but the results are technically correct as far as I could tell. Minor comments: -------------------- - This is obviously a theoretical paper, but It may be useful to say a few things about how much of a role the function smoothness plays in some practical problems? - Just to be clear: the *joint* parameter prior is not explicitly assumed to factorize, only that its marginals are spike-and-slab (equation 8) and that the model probabilities at a given s are uniform, correct?

Reviewer 3



This paper translates some theory of posterior concentration from Bayesian non-parametric regression to deep ReLU networks. This allows for the development of spike-and-slab priors on the weights of the hidden units that both provably concentrates around the true posterior distribution and ought to allow for automatic choice of hidden units. The main proof of posterior concentration follows related work in asymptotic statistics by considering the size of neighborhoods in \norm{.}_n, where n is the number of data points, and considering the limit of n-> \infty. Originality: This seems to be the first paper to apply non-parametric regression techniques to prove posterior concentration properties for spike-and-slab priors on DNNs. This is relatively novel, even if the results are a natural extension of existing work on spike-and-slab variable selection. Significance: The results seem to be a quite natural extension of existing non-parametric regression results; however, this could be a useful addition to the community as it informs potentially more interesting prior choices for researchers applying Bayesian methods to DNNs. Clarity: Presentation wise, the paper seems relatively well structured; however, too much space is spent defining terms and introducing non-parametric regression concepts, resulting in too little space (1.5 pages) being devoted to the new results. Even less space is spent explaining the importance of these results. Additionally, discussion as to how these results can be applied for practitioners is quite lacking. However, the exposition on posterior concentration and spike-and-slab priors is quite nice, well-written, and easy to follow. One suggestion is that the introduction section could be condensed quite a bit, with the relevant references being pushed into the next sections on deep learning and deep ReLU networks. Quality: The proofs in the supplementary material do seem to follow logically, and the comments that I have are mostly minor. The work would be made much stronger by the inclusion of an experiments section showing how these DNNs could be trained. Comments: - A tighter bound on function approximation for deep ReLU networks does exist than is given in Section 5.1, as [2] show that a network of depth k and width O(n/k) exists to approximate any function on n data points in p dimensions. Perhaps that argument could be adapted to include sparsity and thus show a tighter bound in Section 5.1, allowing fixed depth and layers, rather than the requirement that the number of layers grow as a function of data. - One primary weakness in this work is in the lack of experiments using the spike-and-slab theory. It would be very nice to see a) that the spike-and-slab priors discussed could be trained in practice, whether by some sort of Gibbs sampling, MCMC, or by a variant of standard SGD, and b) if an architecture could be adaptively trained using the prior on network width using Equation 18. - It is interesting that the composed deep network suggested by [1] (~180 parameters) performs so much better than the wide shallow network (~8000 parameters). Perhaps the authors can comment on if they would expect this result to hold in the limit of width as the single layer neural network converges to a Gaussian process? - The results seem to hinge on the true generative model, P_f0^n, being in the same class of functions as the model. Could the authors clarify if this is the case, and if not, what would happen to both sparsity and the posterior distribution? - Proof of Theorem 5.1: In Section 3, the authors point to the fact that they must verify all three conditions (Equations 11-13); however, in the proof, they only verify Equations 11 and 12. A quick sentence demonstrating why they do not need to verify Equation 13 would be quite helpful. - Corollary 5.1: In line 274, s_n = n \epsilon_n^2, which has been assumed to approach infinity in the limit (line 183). Thus, it seems like the corollary is just showing that the sparsity level is finite (and thus the number of non-zero parameters is finite), when it should be finite by construction of the neural network. Could the authors comment on this interpretation? Updates after Author Responses: I thank the authors for addressing my mathematical comments, questions about implementations, and comments about paper structuring. These resolve many of my concerns with the paper. Minor Comments/Typos: Line 69: Similarly as dropout, -> Similarly to dropout Line 79: Section 3 defined … -> Section 3 defines (rest of paragraph is in present tense) Equation 7: A verification that this holds for all L>1, p>1 and N>1 would be nice. It is possible to find nonsensical parameter settings that do not satisfy this inequality under those conditions, but I could not find a counter-example to the inequality. Line 168: Potentially (T \choose s)^{-1} instead of 1/ (T \choose s) for clarity. Also applies to other locations. Line 178: The posterior distribution should also include dependence on X (the observed features) and not just Y (the observed classes). Line 201: A citation for “it is commonly agreed …” would be quite helpful. Line 297: We proof… -> We prove… Line 297: There seems to be a missing \Epsilon. log N (…) -> log N \Epsilon (…) Line 307: Where does \epsilon_n/2 come from? It seems like the triangle inequality would give \epsilon_n instead. References: [1] Mhaskar, H., Liao, Q., and Poggio, T. A. When and why are deep networks better than shallow ones? AAAI, 2017. [2] Zhang, C., et al. Understanding Deep Learning Requires Rethinking Generalization, ICLR, 2017. (see particularly Theorem 1 and Lemma 1 in Appendix C.)